# Inferring Attribute Subspaces from Visual Contexts

## Abstract

Recent advances in generative vision-language models have demonstrated remarkable capabilities in image synthesis, captioning, and multi-modal reasoning. Among their most intriguing use cases is in-context learning, the ability to adapt to new tasks from just a few examples. While well-studied in language models, this capability remains underexplored in the visual domain. Motivated by this, we explore how generative vision models can infer and apply visual concepts directly from image sets, without relying on text or labels. We frame this as an *attribute subspace inference* task: given a small set of related images, the model identifies the shared variation and uses it to guide generation from a query image. During training, we use auxiliary groupings to provide weak structural supervision. At inference time, the model receives only unlabeled inputs and must generalize the visual concept based on example images alone. Our approach enables attribute-consistent image generation and contributes a novel direction for nonverbal concept learning in vision.

## 1 Introduction

Generative vision-language models (VLMs) have rapidly advanced in recent years, achieving impressive results in image captioning, visual question answering, and text-conditioned image synthesis (Liu et al., 2023a; Alayrac et al., 2022; Chen et al., 2025). Among their most intriguing use cases is in-context learning: adapting to new tasks or inputs simply by conditioning on multi-modal example prompts, without any parameter updates (Dong et al., 2024). This emergent behavior, particularly well-studied in large language models (LLMs), reflects a core aspect of intelligence: the ability to generalize from a limited context with no additional supervision.

While in-context learning has been widely explored in the textual domain, its visual counterpart remains underdeveloped. Even though recent proprietary VLMs (Fortin et al., 2025; Batifol et al., 2025; Wu et al., 2025) claim to offer multi-modal reasoning capabilities, their ability to understand complex instructions from a visual context remains limited (Figure 2). When faced with instructions they do not understand, these models often collapse to reproducing the input image, generating objects they are biased towards, or exhibiting other degenerate behaviors.

Models with genuine visual in-context learning capabilities should be able to generalize novel concepts from just a few images, reasoning directly over visual input sets without relying on language, labels, or fine-tuning. In this work, we take a step towards this goal by introducing a framework for few-shot attribute discovery from visual context. We formalize this as an *attribute subspace inference* (ASI) task (Figure 1): the model is given a small context set of images that share a common semantic attribute, and it must infer a latent subspace that captures this shared variation. For instance, given a few objects with a common shape, texture, or pose, the model should isolate the relevant factor and apply it to new query images. To teach this capability, we construct training sets using auxiliary information, e.g., groupings derived from the WordNet hierarchy of ImageNet. These groupings act as weak supervision, providing a scaffold that helps the model learn how meaningful visual attributes co-vary across examples. At inference time, it operates solely on unlabeled image sets and a query image to perform attribute discovery and manipulation. This formulation differs from standard text-conditioned generation, few-shot classification, or vision-language pretraining. It also avoids explicit attribute disentanglement or optimization at test time. Instead, our method enables

Figure 1: **Visual in-context learning through attribute subspace inference.** *Left:* Given a context set of images sharing a latent attribute, the model infers a subspace that captures the shared variation. Next, it projects a query image into this subspace. Conditioning on this projected query enables generating new samples that reflect the relevant attribute while discarding unrelated information. The same query produces different outputs under different contexts, showing that the model can isolate and apply attribute-specific structure. *Right:* Additional examples illustrating the expected behavior across different query images.

nonverbal, few-shot concept learning by encouraging a pretrained generative model to reason over the latent subspace structure induced by shared visual context.

We evaluate our method using image synthesis tasks in which attribute subspaces are inferred from a handful of context images and applied to query images to isolate the target attribute. The results show that our model can capture visual attributes, even when those attributes are difficult to verbalize or not represented in standard taxonomies. We summarize our contributions as follows:

- We introduce a new perspective on visual in-context learning, where a model infers and applies latent visual attributes based solely on a small set of related images and a query.
- We propose the *attribute subspace inference* task and develop a training framework that enables generative models to learn this behavior using weak supervisory signals derived from auxiliary grouping information.
- We present qualitative and quantitative results showing that the model can isolate visual factors from the query image and use them to guide attribute-consistent image generation, without relying on textual prompts or class labels.

## 2 RELATED WORK

**Visual In-Context Learning** Visual in-context learning refers to the ability of visual models to adapt to new tasks using only examples provided during inference, without additional training. Although extensively studied in textual domains with large language models (LLMs) (Dong et al., 2024; Brown et al., 2020), visual in-context learning remains relatively under-explored. Recent VLMs have begun addressing this challenge, yet only a handful support support both unified understanding and generation of images, meaning they can reason over multi-modal text-image input and also generate images. BAGEL (Deng et al., 2025) and ILLUME+ (Huang et al., 2025) offer this functionality and currently represent the state-of-the-art in unified image understanding and generation amongst open-source models. Very recently, the largest proprietary models have showcased surprising multi-modal capabilities. Yet, they still fail at correctly solving our proposed ASI task, as shown in Table 1. Other approaches that explicitly target visual in-context learning, such as Visual Prompting via Image Inpainting (Bar et al., 2022), Improv (Xu et al., 2023), sequential prompting (Bai et al., 2024), and Hummingbird (Balazevic et al., 2023), adapt through visual cues or prompts but require explicit task demonstrations in-context.

Our approach differs by removing the reliance on explicit in-context examples. Instead, our model implicitly discovers relevant attributes through provided image sets, closely resembling how humans learn from contextual examples. This framing provides a more natural and flexible mechanism for visually specifying complex, subtle, or otherwise difficult-to-describe attributes, significantly extending the scope and practicality of visual in-context learning.

**Visual Attribute Learning with Explicit Labels** A long line of research has explored learning disentangled image representations without large-scale, per-attribute annotations. Unsupervised

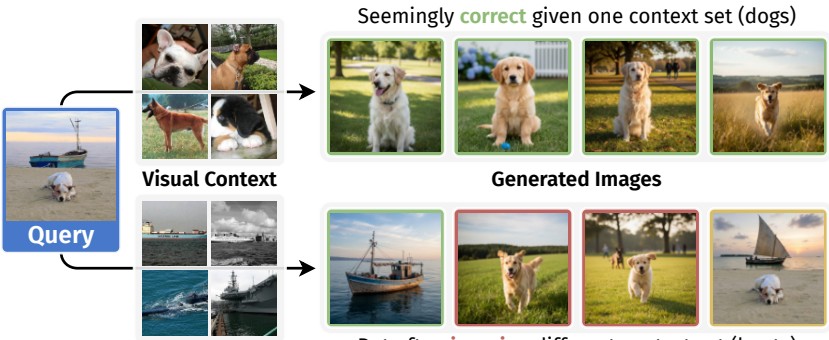

Figure 2: **Typical VLM failure cases:** The model largely ignores the visual context and fixates on a single attribute of the query image, in this case the dog. While the first row may suggest that it has understood the task, the second row reveals a bias toward specific attributes rather than genuine task comprehension.

latent factorization methods like StyleGAN (Karras et al., 2019) and GANalyze (Goetschalckx et al., 2019) reveal that certain latent directions correspond to meaningful variations (e.g., color, style). Other approaches (Higgins et al., 2017; Chen et al., 2018) use regularization or mutual information constraints to discover interpretable factors like pose or lighting, but often require massive datasets or manual inspection. More recently, diffusion models have been analyzed for their capacity to encode structured semantic variations (Jiang et al., 2024; Gandikota et al., 2025), but these methods typically require extensive model finetuning or optimization to extract meaningful attribute directions.

**Measuring Intelligence of Models**   Recent benchmarks like the Abstract Reasoning Corpus (ARC) (Chollet, 2019), CLEVR (Johnson et al., 2017), and physics-based reasoning tests (Motamed et al., 2025) assess model intelligence by evaluating reasoning abilities on visually grounded challenges. However, these approaches typically focus on explicit input-output mappings within well-defined constraints. Our proposed ASI task is uniquely flexible, simulating human cognitive tests by implicitly defining visual attributes through examples. It evaluates a model's capability to generalize from abstract attribute definitions to specific instantiations without explicit input-output pairs, aligning more closely with human visual intelligence assessment.

## 3 METHOD

Our goal is to provide an intuitive and effective mechanism for explicitly instructing a visual model about attributes of interest. We propose an approach framed as an *attribute subspace inference* (ASI) task, where the model learns to identify shared visual attributes within image sets. By explicitly discovering and encoding attributes, we ensure that the model learns semantically meaningful and interpretable feature directions.

### 3.1 DEFINING THE VISUAL ATTRIBUTE SUBSPACE INFERENCE TASK

We frame our learning problem as a visual in-context task, where a model observes a small set of related images and is expected to generalize the underlying visual concept they share. This setup simulates a natural form of visual learning, in which a user conveys a concept by providing example images, without relying on language or labels.

Formally, let an image $\mathbf{x} \in \mathbf{X}$ be associated with a set of visual attributes $A_{\mathbf{x}} \subseteq A$, where $A = \{a_1, a_2, \ldots, a_n\}$ denotes the complete space of possible visual characteristics. Each attribute $a_i$ may have multiple concrete instantiations, such as *car type: sedan*; *surface finish: metallic*; or *color: blue*.

To define an attribute inference task, we construct a *context set* $\mathbf{X}_{\text{set}}$ consisting of several images that share a common (but unlabeled) attribute. For example, the context set might contain images of various dog breeds, objects in bright lighting, or items made of wood. This shared structure encourages the model to infer a latent attribute subspace that captures the variation across the set.

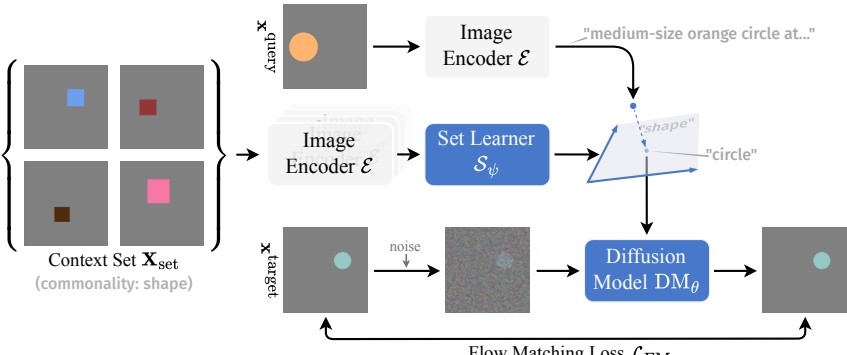

Figure 3: **Complete Model Overview.** Given a set of images that share an attribute (here, shape), the set learner predicts an attribute space that captures the full extent of that visual characteristic. We project the embedded query images in that attribute space to obtain its concrete instantiation (here, circle) and remove all other attributes that cannot be represented in that space. Finally, the diffusion model is conditioned on the projected query to denoise the target image.

The goal is for the model to identify and internalize the relevant visual dimension that unifies these examples.

In order to verify whether the model truly understands and precisely identifies the intended attribute, we introduce two images: a query image $\mathbf{x}^{\text{query}}$ and a target image $\mathbf{x}^{\text{target}}$. Both the query and target images share the same specific instantiation of the attribute (e.g., the same breed of dog). The role of the query image is to concretely instantiate a particular example within the broader attribute category defined by the set, as illustrated in Figure 4. The target image then acts as the correct match or answer, sharing this exact instantiation.

To enable learning of this behavior, we require a mechanism for constructing context sets and corresponding query-target pairs that reflect shared attributes. This relies on auxiliary information that allows us to group images according to some shared semantics. While this introduces a form of indirect supervision, the key idea is to teach the model how to infer structure from sets of images alone. At inference time, the model is presented with only a context set and a query image, and must reason about the relevant visual concept based purely on visual input.

## 3.2 ARCHITECTURE

Our method comprises three components: (1) the *Set Learner*, (2) the *Projection Stage*, and (3) the *Diffusion Model*. See Figure 3 for an illustration. Each component plays a specialized role in our framework to discover, encode, and utilize meaningful visual attributes.

**Set Learner** The Set Learner identifies attribute-specific directions in the feature space. Each image in the input set $\mathbf{X}_{set}$ is first encoded individually using a pretrained Vision Transformer $\mathcal{E}$ (Caron et al., 2021; Oquab et al., 2023; Radford et al., 2021), producing feature token embeddings. The Set Learner itself is a separate Vision Transformer (ViT) that takes the combined embeddings of the entire set of images as input. It reasons across these embeddings jointly to identify the shared attribute and subsequently predicts a set of direction vectors $\mathbf{D}_a = \{\mathbf{d}_1, \mathbf{d}_2, \ldots, \mathbf{d}_k\}$. These directions explicitly represent the subspace corresponding to the shared attribute identified within the set.

**Projection Module** Given a query image $\mathbf{x}^{\text{query}}$, we encode it individually using the same pretrained encoder $\mathcal{E}$, but utilize only the CLS token embedding $\mathbf{e}^{\text{query}}$. To isolate the attribute-specific information identified by the Set Learner, we project this embedding onto the subspace spanned by the attribute directions $\mathbf{d}_i$ through dot products: 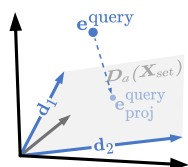

$$s_i = \langle \mathbf{e}^{\text{query}}, \mathbf{d}_i \rangle, \quad \forall \mathbf{d}_i \in \mathbf{D}_a \tag{1}$$

These scalars $\mathbf{s} = \{s_1, s_2, \ldots, s_k\}$ represent the query's attribute instantiation within the broader attribute space. Multiplying each scalar $s_i$ back by the respective direction $\mathbf{d}_i$ and aggregating gives

Table 1: **Performance of current VLMs.** We evaluate the performance of current SOTA VLMs on a simplified version of our task and report accuracy. Even the largest closed-source VLMs are not able to solve our task reliably. Qwen and Flux do not support multiple image inputs. Tiling multiple images into one does not lead to successful task completion.

| Model | Nano Banana (Fortin et al., 2025) | Gemini 2.5 Flash Image (Fortin et al., 2025) | Qwen Image Edit (Wu et al., 2025) | FLUX.1 Kontext Pro (Batifol et al., 2025) | Ours |
|---|---|---|---|---|---|
| **Accuracy** | 46% | 40% | - | - | **93%** |

Figure 4: **Hierarchical Attribute Spaces.** A simplified visualization of our ImageNet hierarchy and attribute space definition. We create a hierarchy of ImageNet classes loosely based on the WordNet hierarchy. We define an attribute space with respect to the next hierarchy level. The $\mathbf{D}_a(\text{animals})$ attribute space differentiates roughly between different types of animals whereas $\mathbf{D}_a(\text{felines})$ is more granular and operates on a lower level of the hierarchy.

a single attribute-conditioned token:

$$\mathbf{e}_{proj}^{\text{query}} = \sum_{i=1}^{k} s_i \mathbf{d}_i \tag{2}$$

This resulting token $\mathbf{e}_{proj}^{\text{query}}$ explicitly captures only the attribute-specific information from the query, discarding unrelated visual details (e.g., background or unrelated objects).

**Diffusion Model**    The final diffusion model $\text{DM}_\theta$ is conditioned on $\mathbf{e}_{proj}^{\text{query}}$ to generate target images $\mathbf{x}^{\text{target}}$. Specifically, $\mathbf{e}_{proj}^{\text{query}}$ is added directly to the timestep embedding of the diffusion model, ensuring strong attribute-conditioned generation.

To train the entire setup, we use the rectified flow objective (Lipman et al., 2023; Albergo et al., 2023; Liu et al., 2023b), which frames generative modeling as matching a continuous vector field. Specifically, given a data distribution $p(\mathbf{x})$ and an initial noise distribution, the diffusion model DM learns a time-dependent vector field $\mathbf{v}(\mathbf{x}, t)$ that transports samples smoothly from noise toward the data manifold. Formally, this involves minimizing the Flow Matching loss:

$$\mathcal{L}_{\text{FM}}(\theta) = \mathbb{E}_{t \sim \mathcal{U}[0,1], \, \mathbf{x}_t^{\text{target}} \sim p_t, \mathbf{e}_{proj}^{\text{query}}} \left[ \left\| \text{DM}_\theta(\mathbf{x}_t^{\text{target}}, t, \mathbf{e}_{proj}^{\text{query}}) - \mathbf{u}(\mathbf{x}_t^{\text{target}}, t) \right\|_2^2 \right] \tag{3}$$

This end-to-end training with a flow matching loss provides a stable objective that is easy to optimize and can work well with smaller batch sizes, which sets it apart from other methods such as contrastive learning that typically requires larger batch sizes to work successfully.

## 4 EXPERIMENTS

We first analyze how well current SOTA VLMs can perform visual in-context learning by measuring their performance on a simplified version of our task (Section 4.1).

Next, we investigate more thoroughly our model's ability for visual in-context learning in two settings using large datasets (Section 4.2). We start by validating our setup in a synthetic setting, allowing for perfect control over the data. Next, we scale our approach to real-world data. To that end, we leverage a generic scheme to assemble context sets and query-target pairs based on the WordNet hierarchy (Miller, 1994).

A correct solution to the Attribute Subspace Inference (ASI) task consists of inferring the relevant attribute from the context set and correctly extracting that attribute from the query image. We

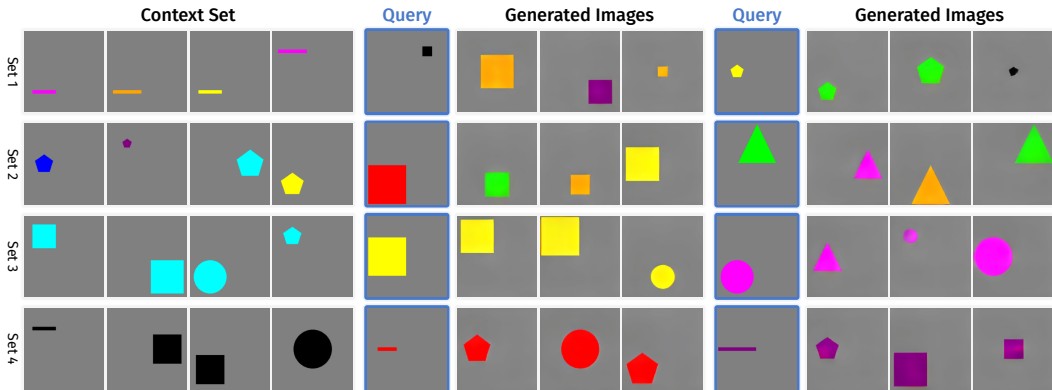

Figure 5: **Visualization of our toy dataset and results.** The left third shows the context set used for obtaining an attribute space, here shape (row 1-2) and color (row 3-4). The middle and right part show three samples for two different query images (left column), respectively. The model learns to correctly identify the attribute specified by the context set and extract it from the context query image $x^{\text{query}}$. Note that only the specified attribute is kept constant while other attributes can vary freely, indicating that the model correctly disregards irrelevant information.

analyze our model's capability to correctly solve ASI tasks from two perspectives. (1) By analyzing if generated images for given context sets and queries pose correct solutions to the ASI task (Section 4.3). (2) By investigating the representation of the queries in the attribute space(Section 4.4).

## 4.1 PERFORMANCE OF CURRENT SOTA VLMS

Since many current SOTA VLMs are closed-source and only accessible via API providers, we perform a small-scale evaluation using the FAL.ai API (fal, 2025) for a simplified version of our task. The dataset consists of an ambiguous query image showing two attributes, and one context set for each of the two attributes visible in the query image. The task for the model is to understand what attribute to extract from the query based on the context set, and generate a variation of the attribute, similar to Figure 1. We prompt the VLM accordingly and manually classify whether the model solved the task correctly. We find that even current SOTA VLMs are not able to solve this task reliably, showing the need for a more principled approach (Table 1). More details can be found in Section C.

## 4.2 DATASETS

In order to train our model and perform a large-scale quantitative evaluation, we introduce two datasets:

**Synthetic Dataset**    To assess whether our model is generally able to reason about sets of images and find the shared attribute in a context set, we construct a synthetic dataset that allows us to precisely obtain images with desired attributes for the context set and the query-target pair. Each image consists of a single object with relevant attributes $A_x = \{\text{shape}, \text{size}, \text{location}, \text{color}\}$. We construct context set, query and target such that one attribute is shared within the context set. Query and target share a potentially different instantiation of that same attribute (see Figure 5).

**Real-World Hierarchical Dataset**    To enable learning of meaningful attribute representations, we employ a hierarchical structure derived from WordNet (Miller, 1994). Each node in this hierarchy corresponds to a visual attribute category (e.g., *animal*), with branches representing increasingly specific attribute values (e.g., *mammal→dog→bulldog*). Sets are constructed at different hierarchy levels to define varying degrees of specificity for attribute encoding. For instance, if the attribute *animal* is chosen, a set might contain mammals, birds, and fish. Given a query image depicting a mammal, the diffusion model should generate other mammals, since that is the next level in the hierarchy. Conversely, selecting a more specific attribute (e.g., *dog*) restricts generation to matching exact breeds.

Table 2: **Quantitative comparison of models on ASI tasks.** Correctly solving an ASI task entails generating correct *and* diverse images. We investigate the models' ability to generalize to sketches from the ImageNet Sketch dataset (Wang et al., 2019) and to unseen classes from ImageNet21k (Deng et al., 2009). We additionally visualize the results for the hierarchy dataset on the right to showcase the trade-off between accuracy and diversity that our model strikes best.

| Method | Dataset | Accuracy (%) ↑ | | Diversity ↑ |
|---|---|---|---|---|
| | | per Attr. | per Val. | |
| Baseline: Copy Query Image | | – | – | 0.47 |
| ILLUME+ 3B (Deng et al., 2025) | Hierarchy | 37.15 | **55.00** | 0.57 |
| BAGEL 7B-MoT (Huang et al., 2025) | | 26.05 | 33.76 | 0.46 |
| Visual Prompting (Bar et al., 2022) | | 26.02 | 45.89 | 0.70 |
| Ours | | **46.34** | 54.46 | **0.81** |
| Visual Prompting (Bar et al., 2022) | Sketch Queries | 23.76 | 44.57 | 0.68 |
| Ours | | **32.60** | **47.11** | **0.72** |
| Visual Prompting (Bar et al., 2022) | Sketch Context | 27.67 | 47.04 | 0.71 |
| Ours | | **41.50** | **52.24** | **0.79** |
| Visual Prompting (Bar et al., 2022) | Sketch C+Q | 25.69 | 44.45 | 0.68 |
| Ours | | **31.90** | **46.47** | **0.71** |
| Visual Prompting (Bar et al., 2022) | 21k Context | 29.07 | 49.95 | 0.76 |
| Ours | | **39.63** | **51.40** | **0.79** |

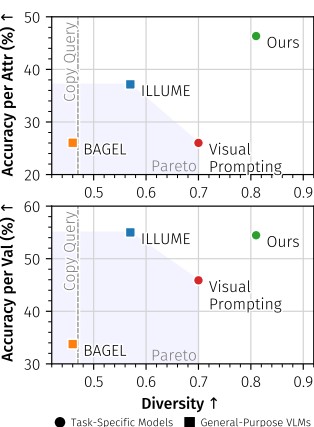

This hierarchical approach addresses a fundamental challenge in attribute learning: constructing meaningful training sets where the relationships between context set, query, and target directly influence the learned representations. While extensive annotations for attribute values would be ideal, such datasets are rare and challenging to create. Our hierarchical structure provides a flexible yet structured framework for attribute subspace inference, where the precise definition of sets directly shapes the model's learned capabilities.

## 4.3 VISUAL IN-CONTEXT LEARNING

If the model can correctly infer the relevant attribute from a context set and extract it from the query, it should generate targets that match the query with regards to the relevant attribute. Thus, we analyze the images generated by our model to determine if the Set Learner has correctly inferred the relevant attribute. In the hierarchical setting, a correct solution entails understanding the hierarchy level and branch from the context set, identifying the query's position within the hierarchy, and generating samples from the appropriate branch. We compare our approach against the Visual Prompting model (Bar et al., 2022), as it is the closest existing method to our approach. Implementation details on how we adapt their method to the proposed ASI task are provided in Section J. Adding to our evaluation of general-purpose VLMs from Section 4.1, we evaluate two state-of-the-art open-source VLMs, (Deng et al., 2025; Huang et al., 2025). Since these general-purpose models have to understand the task in-context, we design three different prompting schemes shown in Table F and ablate their performance in Table E. The VLM results in Table 2 use the best performing prompting scheme.

To quantify performance, we focus on the *animal* subtree of our hierarchy. We chose this subtree for its intuitive structure, as it follows the biological taxonomy, providing a clear hierarchy for evaluation. To quantify both the correctness and diversity of generated images, we report *accuracy* and *diversity* in Table 2. Diversity of generated samples is a crucial indicator of the model's reasoning capabilities, as the trivial solution of copying the query often fulfills the posed task without requiring understanding of the task. This behaviour is punished by our diversity metric.

**Accuracy** We evaluate the model's performance on ASI tasks by measuring the accuracy of the generated images. To assess whether a generated image is in the correct part of the hierarchy, we use a classifier to obtain the class label and locate the predicted class within the hierarchy tree. We provide the mean accuracy averaged over attributes (*per attr*) and averaged over possible attribute instantiations in the query (*per val*). We observe that our model outperforms the Visual Prompting Model model on unseen validation samples from the same hierarchy as used during training, as shown in the upper two rows of Table 2.

**Diversity**   We want the model to demonstrate understanding of the ASI task at hand by not only producing correct but also diverse images. A trivial solution to correctly retrieve an attribute from the query is to simply retrieve all attributes, merely copying the query. As this would not be penalized by a lower accuracy, we introduce a diversity score as a second decisive metric. This score captures the model's ability to generate diverse but correct outputs and penalizes trivial solutions. We define the diversity score as the mean ratio of entropies between generated distributions for a single attribute instantiation $q_{\text{val}}$ projected onto two attribute directions of different abstraction levels:

$$Div = \frac{1}{K \cdot Q} \sum_{q_{\text{val}} \in Q} \sum_{k=1}^{K} \frac{H(q_{\text{val}}, \text{attr}_k^0)}{H(q_{\text{val}}, \text{attr}_k^1) + 1} \tag{4}$$

$Q$ is the number of attribute instantiations for the query image. $K$ denotes the number of attribute pairs where $\text{attr}_k^1$ is a sub-attribute of $\text{attr}_k^0$, i.e. a direct subclass in the hierarchy (e.g. $\text{attr}_k^1 = $ fish and $\text{attr}_k^0 = $ animal ). The entropy

$$H(q_{val}, \text{attr}) = - \sum_{a_i \in M_{\text{attr}}} p(a_i | q_{\text{val}}, \text{attr}) \log(p(a_i | q_{\text{val}}, \text{attr})) \tag{5}$$

is calculated over all attribute instantiations $a_i \in M_{\text{attr}}$ of the higher-level attribute's subtree that the attribute instantiation lies in. $p(a_i | q_{\text{val}}, \text{attr})$ denotes the probability that given a context set for attribute $attr$ and a query with instantiation $q_{val}$, the generated image's attribute $attr$ is instantiated as $a_i$. An illustration of the model's desired behaviour is provided in Figure 4: If we project a *cat* onto the attribute direction *feline*, we want the model to generate diverse cats. If we project that same *cat* onto the attribute direction *animal*, we want the model to generate diverse felines, not only cats. By doing so, the model demonstrates that it has correctly inferred the context set's hierarchy level.

To build intuition for our proposed diversity metric, we evaluate two naive baselines. Replicating the query image results in a diversity score of 0.47 and a near-perfect accuracy, only upper-bounded by the classifier's accuracy. Generating completely random images achieves a diversity score of 0.77, comparable to the values reported in Table 2, but leads to near-zero accuracy. The diversity score's theoretical upper-bound equals 1.87 in our proposed hierarchy setup.

### 4.3.1   GENERALIZATION CAPABILITIES

We evaluate our model's ability to generalize beyond its training distribution by testing it under two challenging settings: (1) using sketch images as context and/or query inputs, and (2) providing context images from previously unseen ImageNet21k classes.

To assess whether the model has overfit to appearance cues or has instead learned to capture semantic content, we use the ImageNet-Sketch dataset (Wang et al., 2019). We construct evaluation setups where the context set, the query, or both are drawn from sketch images. An illustration of this setup is shown in the appendix in Section F. Quantitative results for all combinations of real and sketch images in the context and query are reported in Table 2. While performance decreases relative to settings using only real images, our model maintains strong accuracy and consistently outperforms the Visual Prompting baseline across all configurations.

Additionally, we test if the model can correctly infer relevant attributes given novel attribute instantiations not seen during training. To that end, we construct context sets from ImageNet21k and report performance in the bottom section of Table 2. We observe only slight performance degradation, indicating that our model is able to generalize to unseen attribute instantiations.

### 4.4   ATTRIBUTE SPACES

Since the Set Learner in our architecture predicts an explicit subspace, we can additionally analyze the structure and expressiveness of the attribute space to better understand how the model solves the task. As described in Section 3.2, our method predicts a subspace using the attribute directions inferred by the Set Learner. This approach contrasts with traditional dimensionality reduction methods such as PCA and LDA, which derive subspaces from statistical properties of the data rather than explicitly learning attribute-specific directions. Specifically, we evaluate how the number of examples in the context set influences performance, as larger sets should provide stronger signals for identifying the shared attribute.

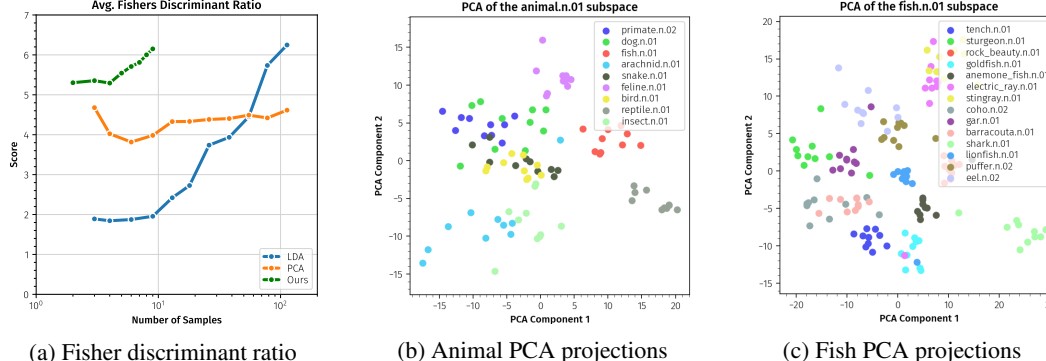

(a) Fisher discriminant ratio      (b) Animal PCA projections      (c) Fish PCA projections

Figure 6: **Attribute Subspace Analysis.** Left: We compare subspaces from our method with PCA and LDA on the same embedding space, using the Fisher Discriminant Ratio (Duda et al., 2000) to assess class separation. Our method achieves higher expressiveness with significantly fewer samples. Middle/Right: Visualization of predicted attribute spaces using the first two principal components. The middle plot shows an `animal` space labeled by the next level of the ImageNet hierarchy; the right shows a `fish` space labeled by class. Our subspaces remain structured and semantically meaningful across different levels of granularity (e.g., `arachnid` appears near `insect` in the `animal` space).

To quantify the structure of the predicted attribute space, we use the Fisher Discriminant Ratio, which assesses class separability by comparing intra-class and inter-class variance. Formally, we compute the between-class scatter matrix $\mathbf{S}_B$ and the within-class scatter matrix $\mathbf{S}_W$, where $N_c$ is the number of elements in class $c$. We then compute the trace ratio $J_{\text{trace}}$ which provides a single scalar measure of how well-separated the attribute representations are:

$$J_{\text{trace}} = \frac{\text{tr}(\mathbf{S_B})}{\text{tr}(\mathbf{S_W})}, \quad \mathbf{S}_B = \sum_{c=1}^{C} N_c(\boldsymbol{\mu}_c - \boldsymbol{\mu})(\boldsymbol{\mu}_c - \boldsymbol{\mu})^\top, \quad \mathbf{S}_W = \sum_{c=1}^{C} \sum_{\mathbf{x}_i \in C_c} (\mathbf{x}_i - \boldsymbol{\mu}_c)(\mathbf{x}_i - \boldsymbol{\mu}_c)^\top. \quad (6)$$

To compare our approach against commonly used dimensionality reduction techniques, we evaluate the expressiveness of our learned subspaces relative to Principal Component Analysis (PCA) and Linear Discriminant Analysis (LDA). To ensure a fair comparison, we average results across multiple context sets and attribute spaces. We encode images using the same pretrained encoder $\mathcal{E}$ and linear projection layer as our method before applying PCA or LDA. While PCA captures general variance across samples, LDA explicitly optimizes for inter-class separability using class labels.

For evaluation, we use the validation split of ImageNet. We randomly sample levels from our hierarchical structure to construct context sets, which in turn define attribute directions. As mentioned earlier, we average results across multiple directions to enable a fair comparison.

We present a quantitative comparison in Figure 6a, showing that our method produces a structured attribute space comparable to LDA but requires an order of magnitude fewer samples. Additionally, we visualize the learned subspace in Figures 6b and 6c, illustrating that a natural structure emerges. As the context set size increases, our method discovers more expressive subspaces. This aligns with intuition: larger sets reduce ambiguity in the ASI task, enabling the model to better approximate the underlying attribute structure.

## 5 CONCLUSION

In this work, we introduced a novel framework for inferring attributes from visual context, formalized as *attribute subspace inference* (ASI) tasks. We propose a scheme to assemble visual context sets from real-world data by leveraging weak semantic groupings of images. We present a training setup that enables learning to infer these attributes from context images and embedding them into a jointly learned low-dimensional attribute subspace. We demonstrated that our model successfully infers relevant attributes from query images and generates diverse outputs that reflect the extracted semantics, even for attributes that are complex, subtle, or hard to verbalize. Our approach outperforms other methods in accuracy and diversity of generated images. We validate the robustness and flexibility of our method, showing its ability to generalize to unseen attributes and modalities such as sketches.

ETHICS STATEMENT

This paper presents work whose goal is to advance the field of generative artificial intelligence with a specific focus on visual understanding and in-context learning. There are many potential societal consequences of our work, none of which we feel must be specifically highlighted here.

REPRODUCIBILITY STATEMENT

We are committed to ensuring the reproducibility of our results. All code will be released with the camera-ready version, including model implementations, training scripts, and the data loader. Detailed implementation information is provided in the main paper and further elaborated in the appendix. We will also release the dataset creation pipelines for both the synthetic dataset and the hierarchical dataset derived from ImageNet, along with the corresponding hierarchy structure. The underlying source dataset (ImageNet) is publicly available. In addition, we will provide the code and data used for the initial VLM accuracy evaluation.

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

# Inferring Attribute Subspaces from Visual Contexts

## Supplementary Material

## A    Limitations and Future Work

Our method relies on auxiliary information to construct context sets and query-target pairs during training. This reflects a fundamental property of the setup: the data itself defines what the model should learn to represent. As a result, the grouping of images carries the semantic structure we want the model to internalize, and the effectiveness of the approach depends on how well these groupings reflect meaningful attributes.

In this paper, we explored two types of auxiliary signals. In a controlled synthetic setting, we had full control over the data generation process. This allowed us to construct batches where the context set shared a specific attribute, the query image instantiated that attribute value, and the target matched the query in that dimension. Such precise supervision is rarely possible outside synthetic data. In our second example, we used the WordNet taxonomy (Miller, 1994) to define groupings over ImageNet classes. In this case, the model learned to capture relationships embedded in the taxonomy hierarchy without relying on explicit attribute labels.

These examples highlight both the generality and the constraint of the approach. Given well-structured groupings, the model can learn to represent and apply a wide range of attributes. However, constructing such groupings at scale remains a practical challenge and a key direction for future work.

One possible path forward is to use unsupervised clustering methods to identify image sets that share latent similarities. For example, subspace clustering (Elhamifar & Vidal, 2013) could be used to discover many overlapping groupings in different embedding dimensions. Context sets and query-target pairs could then be drawn from these clusters, with the assumption that shared structure exists within each. If performed across a broad range of subspaces, the model may learn to generalize across diverse attribute types. The difficulty lies in ensuring that the resulting clusters reflect visually meaningful attributes.

Another direction is to generate attribute annotations automatically using vision-language models. We experimented with this by defining a set of structured attribute-related questions, each with predefined answer sets, and prompting a VLM (Qwen 2.5) (Yang et al., 2025) to label images accordingly. This produced pseudo-attribute labels for ImageNet images, which were then used to form context sets. Although the results showed promise, we found that label quality was often inconsistent, which limited the effectiveness of training. Improving the robustness of such automated annotation pipelines may enable more scalable training in the future.

In summary, our method performs well when given appropriate training data, but future work will need to address how to scale the data construction process. This could include both improving unsupervised discovery of meaningful groupings and refining automated supervision to make the framework more broadly applicable.

## B    General Implementation Details

We train the entire setup end-to-end. The set learner is parameterized as a ViT-L (Dosovitskiy et al., 2021) and the diffusion model as a SiT-L (Albergo et al., 2023). In total this leads to 765 trainable parameters. We train our model on 16 H100 GPUs for 24 hours using 30 query-target image pairs for every set and a global batch size of 80 sets. As the Set Learner only needs to run once per set, we precompute the attribute directions for a set and reuse them for all 30 query-target image pairs. Our attribute space has four directions of dimensionality 256. The image encoder $\mathcal{E}$ is a ViT-L which we initialize with DINOv2 (Darcet et al., 2024; Caron et al., 2021) for faster convergence. We use Adam (Kingma & Ba, 2014) as our optimizer with a learning rate of 1e-4 with linear warmup.

| | Nano Banana(Fortin et al., 2025) | Gemini 2.5 Flash Image(Fortin et al., 2025) | Ours |
|---|---|---|---|
| Replicate Query Image | 4 | 3 | 0 |
| Sample Wrong Attribute | 46 | 40 | 0 |
| Sample Unrelated Image | 4 | 1 | 7 |
| Sample Correct Attribute | 46 | 56 | 93 |
| % Correct | 46% | 56% | **93%** |

Table A: **Error Evaluation.** We roughly classify where the evaluated VLMs fail. In most cases, the model is not able to understand the context set and find the common attribute, leading it to also generate the wrong attribute. In rarer cases, the model simply replicates the query image or produces an unrelated image to the query and context set.

Table B: Performance with noisy context sets. One or two of five context images are replaced with random images from the dataset.

| Set Type | Accuracy (%) ↑ | | Diversity ↑ |
|---|---|---|---|
| | per Attr. | per Val. | |
| clean | 46.34 | 54.46 | 0.811 |
| noisy (1/5 random) | 37.62 | 46.85 | 0.8124 |
| noisy (2/5 random) | 26.01 | 36.42 | 0.9054 |

## C  ADDITIONAL VLM EVALUATION DETAILS

We use the FAl.ai API (fal, 2025) to access closed-source models. Due to the cost of using those APIs, we only generate 100 images per model and manually classify them into the categories shown in Table A. Since these models have no way of knowing the ImageNet hierarchy, we provided a simplified hierarchy in the prompt to make the task more comparable to our main evaluation. The prompt used is: "I will provide you with 5 images. The first four images all contain one shared attribute, i.e., they have something in common. Your goal is to identify this shared attribute and find the corresponding level in the provided hierarchy. Then take a look at the fifth image and find the correct child branch of the selected hierarchy level, i.e., the correct first level under it. Then generate an image showing a random instance from that level. The hierarchy: [hierarchy]". Since we only measure the accuracy of the model, the hierarchy is not strictly required as the model does not need to sample diverse images from the correct hierarchy level, but we found that including the hierarchy increases performance.

## D  LATENT SPACE INTERPRETABILITY

We qualitatively explore how semantically meaningful the predicted subspaces are by interpolating between two data points in a shared subspace. We first select a meaningful attribute space for the interpolation, e.g., feline, project images of cats and tigers in the attribute space, and find an attribute direction by taking the average delta between the embedded tiger and cat points. We then sample points from this direction and conditionally generate images using the diffusion model. In Figure A we show examples of such interpolations.

## E  ROBUSTNESS TO CONTEXT SET QUALITY AND SIZE

We extend the analysis in Sec. 4.3 to test robustness with respect to context set quality and size. The model degrades gracefully under noise and improves with larger context sets. This aligns with the Fisher ratio trend in Fig. 6, which suggests that larger sets reduce ambiguity.

**Sensitivity to context noise**  We replace one or two images out of five with random images from the dataset. As noise increases, accuracy decreases progressively, which indicates that the model uses relevant context effectively while remaining robust to partial corruption. Diversity remains stable or slightly higher, which suggests no collapse under less informative inputs. Results appear in Table B.

| Sample class $a$ | Interpolation $a \rightarrow b$ | Sample class $b$ |
|:---:|:---:|:---:|

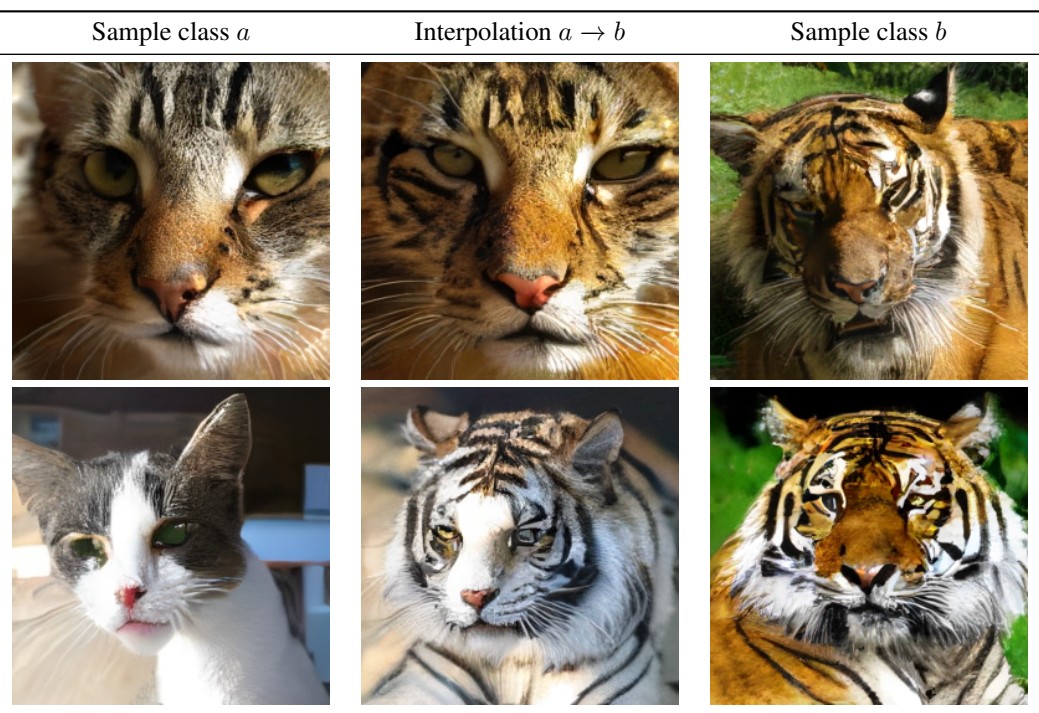

Figure A: Interpolation Examples between two points in a shared subspace.

Table C: Performance across context set sizes. Accuracy improves with additional context, especially in the low-data regime, and diversity increases with size.

| Set Size | Accuracy (%) ↑ | | Diversity ↑ |
|:---:|:---:|:---:|:---:|
| | per Attr. | per Val. | |
| 2 | 44.74 | 48.92 | 0.719 |
| 3 | 45.26 | 53.51 | 0.801 |
| 4 | 46.05 | 54.57 | 0.821 |
| 5 | 46.34 | 54.46 | 0.811 |
| 6 | 46.38 | 54.57 | 0.816 |
| 7 | 46.56 | 54.74 | 0.824 |

**Effect of context set size** We vary the number of context images from two to seven. Accuracy improves steadily with additional context, particularly from two to four examples, and then plateaus. This suggests that small sets already provide useful signal, while larger sets increase reliability for nuanced attribute inference. Diversity also increases with context size. Results appear in Table C.

## F  QUALITATIVE EXAMPLES FOR GENERALIZATION

We provide qualitative examples for the generalization capabilities of our model in Figure B. An extensive quantitative evaluation is provided in the lower part of Table 2.

## G  MULTIPLE SHARED ATTRIBUTES

Real-world scenarios often involve context sets with multiple shared attributes, creating ambiguity about which properties to preserve. To evaluate our model's robustness in such a setting, we conduct controlled experiments using the synthetic datasets.

Figure B: When a query does not fully match the attribute space defined by the context set, it aligns with the closest point on the learned manifold. Although trained only on ImageNet, the model generalizes to sketch inputs by extrapolating semantically. This applies to both query and context images. In the last two rows, we show how different hierarchy levels affect generation: projecting a tiger onto the `feline` space yields tiger-like samples, while projection onto the broader `animal` space results in more generic felines.

Table D: Performance evaluation on synthetic dataset with single and multiple shared attributes. Strict accuracy requires both attributes to match simultaneously.

| Setting | Mean Accuracy (%) ↑ | Mean Entropy ↑ | Random Guessing (%) |
|---|---|---|---|
| 1 shared attribute | 90.96 | 1.88 | 13.85 |
| 2 shared attributes (strict) | 90.57 | 1.77 | 1.87 |

We use our synthetic dataset (Section 4.1) to construct context sets with exactly one or two shared attributes, enabling precise evaluation of the model's behavior under ambiguous conditions. The dataset contains four attributes: shape (5 values), size (7), position (9), and color (10).

**Experimental Setup:** We generate 100 queries and corresponding context sets, sampling 100 outputs per query-context combination (10,000 total images). We measure:

- **Accuracy**: Preservation of shared attribute values using pretrained classifiers ($> 98\%$ validation accuracy)

- **Entropy**: Diversity of non-shared attributes, with bounds from query replication (0) to uniform distribution (2.01)

For the two-attribute setting, we report strict accuracy where a generated image receives 100% accuracy only if both predicted attributes match the ground truth, and 0% otherwise. This evaluation ensures that the model genuinely preserves multiple attributes simultaneously rather than succeeding on individual attributes independently.

The model maintains high accuracy even with two shared attributes while preserving substantial diversity (entropy well above replication baseline). Notably, accuracy remains stable when transitioning from one to two shared attributes, indicating that richer conditioning signals help the diffusion model extract shared properties more precisely without sacrificing diversity.

## H  PROMPT ABLATION ILLUME+

To establish the effectiveness of our approach relative to existing multimodal systems, we conduct a comprehensive evaluation against ILLUME+ (Huang et al., 2025), a state-of-the-art vision-language model that supports both image understanding and generation tasks.

Table E: Performance comparison of three prompting strategies for ILLUME+ versus our method. All metrics are reported higher is better (↑).

| Method | Acc. per Attr (%) ↑ | Acc. per Val (%) ↑ | Diversity ↑ | LPIPS-D ↑ |
|---|---|---|---|---|
| ILLUME+ Prompt #1 | 33.17 | 51.08 | 0.578 | 0.654 |
| ILLUME+ Prompt #2 | 37.15 | **55.00** | 0.568 | 0.631 |
| ILLUME+ Prompt #3 | 33.19 | 50.56 | 0.575 | 0.642 |
| **Ours** | **46.38** | 54.46 | **0.81** | **0.73** |

Since ILLUME+ cannot perform both image understanding (context set analysis) and image generation in a single inference round like our method, we design a two-stage prompting protocol. In the first round, the model analyzes the context set to identify shared attributes. In the second round, it generates a new image based on the identified attribute and the query image. We evaluate three distinct prompting strategies to ensure a fair comparison and follow our standard evaluation protocol described in Section 4.3.

Table E presents the quantitative results comparing different prompting strategies for ILLUME+ against our method. Our approach strikes a significantly better trade-off across evaluation metrics compared to all prompting variants. The best-performing prompting strategy (Prompt #2) achieves 37.15% accuracy per attribute, 55.00% accuracy per value, and a diversity of 0.568, while our method achieves 46.38%, 54.46%, and 0.81, respectively. We detail the three prompting strategies employed for ILLUME+ in Table F.

Our results reveal several key insights. First, prompt optimization significantly impacts VLM performance, with Prompt #2 outperforming Prompt #3 substantially in attribute accuracy. However, even the best-performing prompt falls substantially short of our method's performance overall. Second, the two-stage nature of the VLM approach introduces potential error propagation, where mistakes in attribute identification compound during image generation. Finally, our end-to-end approach demonstrates superior capability in both understanding shared attributes and generating coherent images that preserve these attributes while introducing appropriate variations.

## I  FEW-SHOT CLASSIFICATION USING ATTRIBUTE SUBSPACES

The ASI framework naturally extends to various downstream applications through its inference-time adaptability. We demonstrate this capability through a few-shot classification experiment. We evaluate our method's transferability by applying a model trained on the ImageNet hierarchy to few-shot classification on the Caltech-UCSD Birds (CUB) dataset (Wah et al., 2011). A key advantage of ASI is its ability to adapt the encoder at inference time by defining context sets without requiring additional training or class information.

Our experimental setup uses context sets of five randomly selected bird images to define a bird-specific attribute subspace. We then perform 1-shot 5-way classification by projecting CUB test images into this learned attribute space. This approach achieves an accuracy of **72.28%**.

For comparison, we establish a DINO baseline using an equivalent experimental setup. We create a DINO subspace by embedding the same context images and applying PCA with four components (matching our number of attribute directions). Few-shot classification in this DINO subspace yields a baseline accuracy of **71.78%**. Our method outperforms this strong baseline despite being trained on ImageNet at a much smaller scale than DINO, demonstrating the effectiveness of our attribute-based representation learning.

## J  VISUAL PROMPTING IMPLEMENTATION DETAILS

To ensure compatibility with (Bar et al., 2022), we cast the ASI task as a 3x2 grid image that encompasses a 2x2 context set at the top, a query image on the bottom left and the denoising target image on the bottom right. We train a flow matching model to denoise the bottom right part of this 3x2 grid. To enable a fair comparison to our main model, the flow matching model is parameterized as

Table F: Prompting strategies used for ILLUME+ evaluation.

| Strategy | Round 1: Attribute Identification | Round 2: Image Generation |
|---|---|---|
| **#1** | *Text:* "I provided you with four images and a class hierarchy below. Find the hierarchy level that best matches the commonality / shared attribute in the four images. Answer only with the name of the hierarchy level, nothing else. The hierarchy: [hierarchy]" 
 *Input:* Context set images | *Text:* "I provided you with one image and a class hierarchy below. Find out what child of the [extracted_feature] hierarchy level best matches the content of the provided image. Generate an image showing a random instance from that level. If there are no children under the selected hierarchy level, you can generate an image from the same class of the given image. 
 The hierarchy: [hierarchy]" 
 *Input:* Query image |
| **#2** | *Text:* "Here are four images and a taxonomy. Identify the hierarchy level that captures the common concept present in all four images. Answer only with the name of the hierarchy level, nothing else. The hierarchy: [hierarchy]" 
 *Input:* Context set images | *Text:* "I will give you one image along with a hierarchy of categories. Identify which child node of the hierarchy level [extracted_feature] best describes the image. After identifying it, generate a new image showing a random example from that node. If there are no children under the selected hierarchy level, you can generate an image from the same class of the given image. 
 The hierarchy: [hierarchy]" 
 *Input:* Query image |
| **#3** | *Text:* "I am giving you four example images and a hierarchy of categories. Find the hierarchy level that represents the feature or class they all have in common. Answer only with the name of the hierarchy level, nothing else. 
 The hierarchy: [hierarchy]" 
 *Input:* Context set images | *Text:* "Analyze the given image in the context of the following hierarchy. Pick the child under the hierarchy level [extracted_feature] that best matches this image. Then produce an image showing another random instance from that same hierarchy level. If there are no children under the selected hierarchy level, you can generate an image from the same class of the given image. 
 The hierarchy: [hierarchy]" 
 *Input:* Query image |

a SiT-L (Albergo et al., 2023). Similar to our main model, the context and query images from the 3x2 grid are individually encoded by a ViT-L image encoder $\mathcal{E}$ which we initialize with DINOv2 (Darcet et al., 2024; Caron et al., 2021) for faster convergence. We train the model on 16 H100 GPUs for 24 hours.

## K  ADDITIONAL QUALITATIVE EXAMPLES

We show additional qualitative examples from our hierarchical Imagenet model in Figure C.

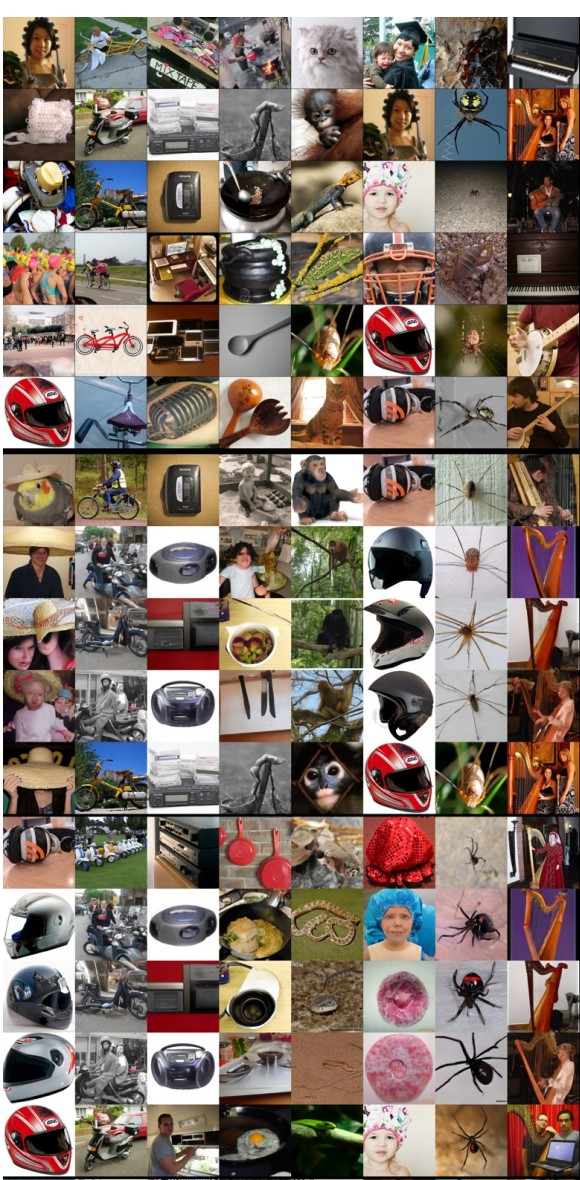

Figure C: More samples from our model. The upper third shows the context sets (column-wise) consisting of five images each. The following two sections are organized in the following way: query, three samples, and target. This visualization intends to provide a better intuition of how the hierarchy is structured and how we compose sets, queries, and targets.

