# OpenReview forum: "Inferring Attribute Subspaces from Visual Contexts"
_ICLR.cc/2026/Conference — ICLR 2026 Conference Withdrawn Submission_

### Official Review · Reviewer_5TBq · 2025-10-31

**Soundness:** 3
**Presentation:** 2
**Contribution:** 2
**Rating:** 4
**Confidence:** 4

**Summary:**

The paper introduces a framework for Attribute Subspace Inference (ASI) which is a formulation of visual in-context learning that allows generative vision models to infer shared visual attributes directly from sets of unlabeled images. Given a small context set of related images and a query image, the model identifies a latent subspace capturing the shared attribute (e.g., shape, texture, species) and applies it to the query for attribute-consistent image generation.

The method uses:
1. Set Learner (ViT) to infer attribute-specific directions from the context set,
2. Projection Module to project query embeddings into that attribute subspace, and
3. Diffusion Model trained with flow-matching loss to generate corresponding outputs.

Experiments show that the model outperforms prior visual prompting methods (e.g., ILLUME+, BAGEL) in both accuracy and diversity metrics, and generalizes to sketches and unseen ImageNet classes.

**Strengths:**

1. The methodology is conceptually clean and easy to follow the set learner infers shared attribute directions, the projection isolates relevant features, and the diffusion model uses them for generation.
2. By focusing on inferring shared visual attributes directly from image sets without text or explicit labels the paper explores a timely and important direction in generative AI. The given results though limited in scope, does show that visual examples alone can guide generation.

**Weaknesses:**

1. The approach requires pre-defined groupings (e.g., WordNet hierarchy) to create context sets, limiting scalability beyond well-structured datasets.
2. The paper combines three existing ideas of attribute subspace learning, diffusion-based conditioning, and few-shot in-context inference into a single framework. Which is not a problem but while each component is individually meaningful, the paper does not clearly articulate why this specific combination is necessary or what unique advantage emerges from their interaction beyond empirical gains. The rationale for how diffusion modeling, subspace reasoning, and in-context inference conceptually reinforce each other is not fully developed; they appear more as compatible modules than as parts of a unified theoretical insight. Strengthening the discussion on why these three elements must coexist rather than merely co-function would make the contribution more compelling.
3. Similar to my previous point  (subspaces + diffusion + in-context inference) are combined coherently. The components could arguably be swapped with alternatives (e.g., GAN, autoencoder, or nonlinear projector) without changing the premise according to my understanding.
4. The projection stage assumes that semantic attributes correspond to linearly separable directions in the ViT embedding space, as evidenced by the use of simple dot-product projections to extract attribute representations. However, the paper provides no theoretical or empirical justification that such linearity holds for complex or abstract visual attributes. Without analysis of whether these attribute directions are truly orthogonal or disentangled, it remains unclear whether the learned subspaces capture meaningful semantics or merely reflect dominant visual correlations within the context set.
5. The conditioning mechanism in the diffusion model is limited to a simple additive injection of the projected query embedding into the timestep embedding. The paper does not analyze whether this approach effectively guides generation compared to established conditioning strategies such as cross-attention, concatenation, or FiLM modulation. Furthermore, the relationship between the learned attribute subspace and the diffusion model’s latent space is not theoretically or empirically examined, making it unclear whether the observed performance stems from meaningful conditioning or from coincidental feature alignment. The choice of flow-matching loss for ''stability'' as mentioned in the paper is also untested.

**Questions:**

1. Without an auxiliary consistency or reconstruction term, the model might learn correlations that satisfy the flow-matching loss but not necessarily correspond to interpretable attributes. Did the authors consider adding a semantic or reconstruction constraint (e.g., contrastive or cycle-consistency loss) to ensure the learned subspaces correspond to meaningful attributes rather than incidental correlations?

2. Given that training supervision comes from hierarchical groupings, is it possible to verify that the model learns semantically meaningful features instead of trivial correlations like color or texture?

---

### Official Review · Reviewer_uBFZ · 2025-11-01

**Soundness:** 2
**Presentation:** 2
**Contribution:** 2
**Rating:** 4
**Confidence:** 4

**Summary:**

This paper introduces a form of visual in-context learning where the generator infers and applies visual concepts based on a visual context set and then applies the attribute to the query (ASI, Attribute Subspace Inference task). The model is first tasked with identifying the shared latent attribute variation from the context set, and then uses it to project a query image into that inferred attribute subspace. Then the resulting token is used to condition a diffusion model generation to get the final output image. The goal of the paper is to facilitate nonverbal few-shot visual concept learning and attribute-consistent image synthesis

**Strengths:**

-  **Concept Learning from reference images**: this paper provides an interesting framework for learning visual concepts from purely from image sets, without requiring textual prompts or explicit labels. This is interesting as not all visual concept are verbalizable or explicitly explainable
- **Robustness and some generalization**: The model maintains strong performance and degrades reasonably even when the context sets are noisy (e.g., 1 or 2 out of 5 images replaced randomly). It generalizes effectively to unseen attribute instantiations (from ImageNet21k) and different modalities (sketches) - although one out of two OOD examples shown in Figure B is incorrect (the generated image is not a Labrador).
- **Improved Performance:** Achieves high accuracy (93%) on the ASI task, reliably outperforming current state-of-the-art closed-source and open-source Visual Language Models (VLMs).

**Weaknesses:**

- **Ambiguity of context–query relationship:** In some visualizations (e.g., Fig. 5), it remains unclear what the model is actually inferring from the context beyond basic object consistency. The generation sometimes introduces attributes not present in the context set (e.g., unseen colors or poses). It would help to clarify how the model determines which transformations are permissible—are these learned priors from the training data or genuinely inferred from the set?
- **Hierarchical assumption not intuitive:** The paper claims that if the context set depicts “animal” and the query depicts a “mammal,” the model should generate other mammals, and if the context set depicts a “dog,” it should generate the same breed. However, this hierarchical mapping seems externally imposed by the dataset design (WordNet grouping / problem setup) rather than naturally emergent from context understanding. Without textual or semantic supervision, it is not clear how the model would learn to map from “context set → hierarchy level → query instantiation” purely from visual examples. This may also attribute to extremely poor performance of VLMs, as these tasks tested in the paper are not intuitive, even to a human (inferring the particular structure of this task from images only).
- **Interpretability and scalability trade-off:** While the attribute subspace design enhances interpretability, it may limit scalability to more complex compositional concepts (e.g., multiple interacting attributes). The model currently operates in a low-dimensional setting and has only been tested on single-attribute or low-ambiguity contexts.
- **Training data dependency:** The method’s reliance on auxiliary grouping (e.g., WordNet hierarchy) introduces a structural prior that effectively defines the learned “attribute space.” As acknowledged in the appendix, the approach’s success depends heavily on how these groupings are curated, which may limit scalability to unstructured or open-domain settings.

**Questions:**

1. **Context inference ambiguity**: In several examples, the generated output includes features (colors, poses) not present in the context set. Is the model extrapolating based on priors from the pretraining data? How does it understand the allowable levels of changes along certain dimensions? Could you quantify how much the generation is influenced by dataset priors versus inferred context?
2. **Hierarchy supervision:**  Since the attribute hierarchy (e.g., mammal → dog → breed) is built from WordNet during training, does the model rely on this structure during inference as well, or can it infer hierarchical relations from unseen sets without such scaffolding?
3. **OOD task analysis**:  You report results on sketches and ImageNet21k classes. Could you provide qualitative examples or a failure analysis for these OOD cases to better illustrate the limits of semantic generalization?

---

### Official Review · Reviewer_gboo · 2025-11-01

**Soundness:** 3
**Presentation:** 3
**Contribution:** 2
**Rating:** 4
**Confidence:** 2

**Summary:**

This paper introduces Attribute Subspace Inference (ASI), a task where a model infers a shared visual concept from a small set of unlabeled images and applies it to guide the generation of a new image from a query. The proposed method uses a "Set Learner" module to predict a low-dimensional subspace representing the shared attribute. The query image's embedding is then projected onto this subspace, and the result is used to condition a diffusion model for image generation. The model is trained using weak supervision derived from pre-existing semantic hierarchies, such as WordNet, without needing explicit attribute labels at inference time.

**Strengths:**

1. The ASI task is a creative and valuable contribution. It proposes an intuitive, non-verbal mechanism for concept learning and conditional generation that moves beyond standard text or class-label conditioning.

2. The architecture, which combines a dedicated module for subspace inference with a standard diffusion model, is logical and well-suited for the task. The end-to-end training with a flow matching loss is a solid and modern choice.

3. Within the confines of the experimental setup, the model demonstrates impressive performance. It significantly outperforms large, general-purpose VLMs on the ASI task and shows robust generalization to out-of-distribution data like sketches and unseen classes from the same hierarchy.

**Weaknesses:**

1. A key challenge in visual reasoning is disambiguating which shared attribute is relevant in a given context (e.g., 'redness' vs. 'car-ness'). The paper's methodology sidesteps this entirely. By constructing training batches where each context set corresponds to a single, unambiguous ground-truth attribute from the hierarchy, the model is never forced to learn how to resolve ambiguity. The failure of SOTA VLMs shown in Figure 2 is largely due to this ambiguity, yet the proposed model's success is predicated on a training setup that removes it.

2. The model is not truly discovering latent attributes from scratch; rather, it is learning a supervised mapping from pre-defined image groupings (provided by the WordNet hierarchy) to a latent subspace. The semantic structure is given a priori by the dataset's construction, not inferred by the model. This fundamentally constrains the model to only "discover" attributes that were explicitly encoded into the training data's structure, weakening the central claim of nonverbal concept learning.

3. While the poor performance of general-purpose VLMs successfully motivates the ASI task, the comparison isn't entirely fair. The proposed model is trained specifically on the ASI task structure, whereas the VLMs are tasked with solving it in a zero-shot, in-context manner. The results demonstrate that specialized training is superior to zero-shot prompting for this novel task, which is an expected outcome. The framing could be nuanced to reflect this, presenting it less as a failure of SOTA models and more as evidence for the necessity of this new training paradigm.

**Questions:**

I would appreciate the authors' feedback on the weaknesses.

---

### Official Review · Reviewer_BDP5 · 2025-11-01

**Soundness:** 2
**Presentation:** 2
**Contribution:** 2
**Rating:** 2
**Confidence:** 4

**Summary:**

This paper presents an approach to identify attribute subspace using visual in-context examples for image generation. A ViT is used to identify the shared attribute between the visual examples to find the feature directions. The query image is used to find a more refined feature representation as it shares the features of the target image. The feature representation of the query image is projected to the feature directions of the visual examples which is then used as conditioning for the diffusion model.
Experiments are performed on toy dataset, sketches dataset and the ImageNet Context21K dataset.

**Strengths:**

The qualitative examples show that for simple toy datasets the approach can identify the relevant attributes.
The approach of attribute subspace learning provides high diversity samples with less number of samples as is shown in the Fisher Discriminant ratio.

**Weaknesses:**

1. The number of attributed studied in the paper are limited. In the qualitative study the work concentrates on shapes without any background information.
2. Comparison to papers on disentanglement: The scope and comparison is performed with VLMs and visual prompting papers. It is not clear if the paper is positioned correctly with respect to the work on disentanglement.
3. The computational overhead incurred
4. Generalization: Since the diffusion weights are updated for the new conditioning mechanism, how does the approach generalize across images or datasets.
5. The set learner : The procedure to form the attribute space is not provided.

**Questions:**

1. Attribute Coverage: Why does the qualitative study focus primarily on shape attributes without considering background or other visual factors? Could the approach handle more complex attributes beyond shape?
2. How does your method compare to existing work on disentangled representation learning? Why were such methods not included in the experimental comparison, and how do you position your contribution relative to that literature?
3. What is the additional computational cost introduced by the Set Learner and projection mechanism compared to standard diffusion inference?
4. I the section on generalization what is the reason that a trained model with specific directions generalizes to arbitrary subspaces.
5. Could you provide more details on how the Set Learner constructs the attribute subspace? What is the learning criterea?

---

### Note · Authors · 2025-11-14

I have read and agree with the venue's withdrawal policy on behalf of myself and my co-authors.